# Assessment of DNA/RNA Defend Pro: An Inactivating Sample Collection Buffer for Enhanced Stability, Extraction-Free PCR, and Rapid Antigen Testing of Nasopharyngeal Swab Samples

**DOI:** 10.3390/ijms25169097

**Published:** 2024-08-22

**Authors:** Mikhail Claeys, Saif Al Obaidi, Karen Bruyland, Ilse Vandecandelaere, Jo Vandesompele

**Affiliations:** 1InActiv Blue, Industriepark Oost 2A, 8730 Beernem, Belgium; mikhail.claeys@inactivblue.com (M.C.); saif.al.obaidi@student.howest.be (S.A.O.); 2Campus Brugge Station, Howest University of Applied Sciences, Rijselstraat 5, 8200 Brugge, Belgium; 3Medisch Labo Bruyland, Beneluxpark 2, 8500 Kortrijk, Belgiumilse.vandecandelaere@bruyland.be (I.V.); 4Department of Biomolecular Medicine, Ghent University, Corneel Heymanslaan 10, 9000 Gent, Belgium

**Keywords:** nucleic acid amplification, rapid antigen testing, SARS-CoV-2, influenza, RSV, inactivating buffer, direct RT-qPCR, extraction-free PCR, nasopharyngeal

## Abstract

This study comprehensively evaluated the DNA/RNA Defend Pro (DRDP) sample collection buffer, designed to inactivate and stabilize patient samples. The primary objectives were to assess DRDP’s efficacy in ensuring sample stability, facilitating extraction-free polymerase chain reaction (PCR), and ensuring compatibility with rapid antigen testing (RAT). Ninety-five diagnostic nasopharyngeal swab samples tested for influenza virus (influenza A), respiratory syncytial virus (RSV A), and/or severe acute respiratory syndrome coronavirus 2 (SARS-CoV-2) were 10-fold diluted with DRDP and anonymized. Initial characterization and retesting of these samples using cobas Liat confirmed 88 samples as positive, validating the presence of viral targets. Results from rapid antigen testing showed lower sensitivity compared to nucleic acid amplification testing (NAAT) but maintained perfect specificity, with 40 out of 88 positive samples by cobas Liat also testing positive for RAT. Direct RT-qPCR of DRDP-diluted samples demonstrated robust compatibility, with 72 out of 88 samples positive for cobas Liat also testing positive by direct RT-qPCR. Non-concordant results could be explained by the 200-fold lower input of extraction-free NAAT. Stability testing involved incubating 31 positive samples at 4 °C, 20 °C, and 37 °C for 7 days, with extraction-free NAAT. DRDP guaranteed viral RNA stability at all temperatures for influenza A, SARS-CoV-2, and RSV A, showing stability up to 7 days at 4 °C. In conclusion, DRDP is an effective stabilizing medium compatible with direct RT-qPCR and rapid antigen testing and shows great potential for optimizing diagnostic processes, particularly in resource-limited or time-sensitive scenarios.

## 1. Introduction

In the realm of diagnostic procedures, preanalytical steps play a pivotal role in ensuring the accuracy of results. The stability of analytes over various durations and under diverse temperature conditions is a critical consideration because it protects against the risk of false negatives. Equally important is the inactivation of pathogens at the site of collection, which is essential for ensuring safety during transport and upon arrival at the test laboratory.

Nucleic acid amplification testing (NAAT) often involves the purification of nucleic acids from the sample matrix to eliminate potential inhibitors and potentially concentrate the analyte. While this enhances analytical sensitivity, it comes at the cost of increased expenses and turnaround time, potentially creating bottlenecks when the test capacity is limited. Consequently, alternative protocols, such as extraction-free or direct PCR or sample pooling strategies, have been developed to alleviate these concerns [1,2].

In instances where the reference NAAT testing capacity is constrained, or rapid results are imperative, a triage testing strategy may be employed [3]. This involves first conducting a rapid antigen test (RAT), followed by a reference NAAT (either or not of only the negatives). This approach aims to enhance the diagnostic yield (by increasing the test population) and/or to lower the number (and costs) of reference NAAT needed. Ideally, the same collection buffer is suitable for both RAT and NAAT; however, compatibility issues may arise.

While procedures exist for extraction-free NAAT, either or not after heat-inactivation of the sample at the laboratory [4,5,6,7,8], to our knowledge, no studies have reported on a sample collection medium with inherent pathogen inactivation properties that preserves RNA and DNA at elevated temperatures for several days and allows both (extraction-free) NAAT and RAT. Indeed, most media that inactivate pathogens (e.g., Primestore, eNAT, InActiv Blue, DNA/RNA Shield) [9,10,11] also denature proteins and antigens and are incompatible with extraction-free NAAT or antigen testing.

Filling this gap, InActiv Blue has developed a novel sample collection and transport buffer known as DNA/RNA Defend Pro (DRDP) [12], specifically designed as a stabilizing medium that inactivates viruses. DRDP is compatible with both (extraction-free) NAAT and RAT.

This retrospective paired study design aims to comprehensively evaluate DRDP as a stabilizing medium for upper respiratory swab samples. In addition, this investigation will assess its compatibility with antigen tests and its suitability for use in direct RT-qPCR applications. We aim to contribute valuable insights into the efficacy and versatility of DRDP in enhancing the reliability and efficiency of diagnostic processes.

## 2. Results

Of the 95 positive samples according to the diagnostic test in Medisch Labo Bruyland (MBL), 88 (93%) also tested positive upon retesting 0.5 mL of a 10-fold dilution using cobas Liat in the InActiv Blue laboratory. As the samples were diluted and stored at varying temperatures and for different durations, a retest was expected to not detect the samples with the lowest signals.

### 2.1. Rapid Antigen Testing (RAT)

Of the 88 samples that tested positive for cobas Liat, 40 (45%) were positive for RAT (Table 1). The seven cobas Liat-negative samples were also negative for rapid antigen testing. No false-positive results were obtained using the RAT. Representative examples of the RAT results can be found in Appendix A.

As expected, samples with a negative RAT result had a significantly higher cobas Liat Cq value (low abundance of viral target), confirming the superior analytical sensitivity of the molecular test (*p* = 1.53 × 10^−6^, 8.53 × 10^−9^, and 3.11 × 10^−4^ for SARS-CoV-2, influenza A, and RSV A, respectively) (Figure 1).

### 2.2. Direct RT-qPCR

Of the 88 samples positive for cobas Liat, 72 (82%) were positive by direct RT-qPCR (Table 2). The seven samples that tested negative for cobas Liat also tested negative by direct RT-qPCR. No false-positive results were obtained by direct RT-qPCR.

As expected, samples with a negative direct RT-qPCR result had a significantly higher cobas Liat Cq value (low abundance of viral target, *p* = 2.97 × 10^−5^, 4.69 × 10^−7^, and 0.038 for SARS-CoV-2, influenza A, and RSV A, respectively), in line with the higher sensitivity because of a 200-fold higher input in cobas Liat testing compared to direct RT-qPCR (200 µL vs. 1 µL) (Figure 2).

For all viral targets, a good to excellent correlation is observed between the direct RT-qPCR test and the CE-IVDR cobas Liat test Spearman r = 0.940 (*p* = 1.10 × 10^−6^), Spearman r = 0.916 (*p* = 2.20 × 10^−16^), Spearman r = 0.791 (*p* = 2.08 × 10^−3^) for SARS-CoV-2, influenza A, and RSV A, respectively) (Table 3, Figure 3). The mean differences in Cq values between the direct RT-qPCR and cobas Liat were closely in line with the approximately 200-fold difference in input (log_2_(200) = 7.64), also considering that different primer and probe sequences and PCR reaction conditions were used.

### 2.3. RNA Stability

Of the 72 samples that tested positive by direct RT-qPCR and cobas Liat, a semi-random subset of 31 (43%) was selected for stability testing (see Section 4). These samples were divided into three 100 µL aliquots and subjected to incubation at 4 °C, 20 °C, or 37 °C. Direct RT-qPCR measurements were performed on incubation days 0, 1, 3, and 7.

A target was defined as detected if the Cq was <40. All targets in all samples were detected on day 7 at all temperatures (Table 4). For target RSV A, one sample was not detected on day 3 but was detected again on day 7. This sample (ID 24-0013) had a high Cq on day 7 (37.07).

For stability based on the mean Cq of all samples within a target, the cut-off was set at 2 Cq per the instructions of use of DRDP. If the delta-Cq of the time points exceeded two compared to time point 0, it was determined to be no longer stable.

Influenza samples remained stable for up to 7 days at all tested temperatures. SARS-CoV-2 and RSV A samples remained stable for up to 3 days at all tested temperatures and for up to 7 days at 4 °C (Table 5).

## 3. Discussion and Conclusions

DNA/RNA Defend Pro (DRDP) is a unique sample collection and transport buffer that inactivates pathogens [12], enables triage and reflex testing using rapid antigen tests (RAT), stabilizes viral RNA for several days at elevated temperatures, and can reduce costs by not necessarily requiring nucleic acid purification prior to nucleic acid amplification testing (NAAT).

Collecting the patient swab sample directly in DRDP would have been the preferred method for performance evaluation. However, to minimize patient discomfort, anonymized nasopharyngeal swab samples provided by Medisch Labo Bruyland were 10-fold diluted in DRDP, as in a previous study [13].

DRDP buffer preserves antigens in their native state, as evidenced by the perfect correlation between the qualitative RAT and the quantitative values of NAAT. As expected, RAT displays lower sensitivity, as this method requires a substantial number of viral antigens for a positive result. Detection of all positive samples by RAT under a certain CE-IVDR extraction-based NAAT Cq value strongly suggests that the decreased sensitivity is attributed to the technique, that DRDP is compatible with RAT, and that antigens are preserved in the inactivation buffer.

Compared to a CE-IVDR NAAT with excellent analytical sensitivity incorporating nucleic acid extraction and purification [14], a somewhat lower sensitivity of direct (or extraction-free) NAAT was observed in our study. As much less sample volume is used in the direct NAAT, this lower sensitivity is expected. The sensitivity could be increased by using an RT-qPCR kit that is even more tolerable to inhibitors, allowing more than 10% buffer input and/or performing PCR in larger volumes with more sample input (e.g., 10 µL sample in a 50 µL PCR reaction, a 10-fold increase compared to the current procedure). The detection of all positive samples by direct NAAT under a certain CE-IVDR extraction-based NAAT Cq value supports the notion that decreased sensitivity is attributed to the dilution factor and that DRDP is perfectly compatible with direct NAAT. While several studies have indicated the suitability of extraction-free SARS-CoV-2 molecular testing, the analytical sensitivity compared to an extraction-based procedure ranged from 80% to 100% [15,16,17]. Importantly, this is partly related to the magnitude of the concentration effect in the extraction method: some procedures only purify nucleic acids (i.e., 200 µL sample input in 200 µL elution buffer), while others purify and concentrate (i.e., 400 µL sample input in 50 µL elution buffer, an 8-fold enrichment).

Finally, viral RNA in swab samples stabilized in DRDP shows stability for several days. Of note is that the stability of viral RNA in DRDP at higher temperatures seems to be virus-dependent. Qualitative presence/absence and quantitative Cq data show the stability of viral RNA from nasopharyngeal swabs in DRDP between 3 and 7 days at all temperatures but in a target-dependent manner. For longer stability, storage at 4 °C is recommended (21 days according to DRDP’s Instructions for Use [18]).

While our results are promising and useful when faced with a pandemic-like scenario of resource constraints or when ultimate sensitivity is not warranted (e.g., to differentiate infectious patients from those with high Cq values that are no longer infectious), more work is needed to further explore the workflow elements that contribute to the analytical sensitivity of direct NAAT using DRDP. Different reverse transcriptase and Taq DNA polymerase enzymes and buffers could be explored, as well as primer/probe sequences and concentrations, increased reaction volumes, or different thermocycling conditions (e.g., incorporating the RT step as part of the PCR assay setup at room temperature [19]). Finally, it remains to be determined whether using another sample matrix like saliva would provide similar results in direct NAAT.

In conclusion, this study demonstrated that DNA/RNA Defend Pro (DRDP) is a highly effective sample collection and stabilization buffer for nasopharyngeal swabs. It ensures the inactivation of pathogens while preserving the integrity of viral RNA at various temperatures for several days. DRDP’s compatibility with both direct RT-qPCR and rapid antigen testing underscores its versatility and utility in different diagnostic settings. Despite the slightly lower sensitivity of extraction-free NAAT compared to traditional methods, the buffer’s ability to maintain viral RNA stability and facilitate accurate antigen detection makes it a valuable tool, especially in resource-limited or time-sensitive scenarios. This study’s findings highlight DRDP’s potential to streamline diagnostic processes, reduce costs, and improve safety in clinical and laboratory environments, paving the way for more efficient and reliable infectious disease testing.

## 4. Materials and Methods

### 4.1. Sample Collection

From December 2023 to February 2024, left-over nasopharyngeal swab samples from Medisch Labo Bruyland (MLB) were set aside at 4 °C or −20 °C for one day up to 2 weeks. The study of stabilizing buffers for the transport and storage of human body specimens was approved by the Ethics Committee of the University Hospital Antwerp (ID 5931). A total of 95 samples were stored that had tested positive for influenza A, RSV A, and/or SARS-CoV-2.

Diagnostic swabs were transported in PBS or Amies buffer to MLB. To ensure that results obtained in this study reflect the capabilities of DNA/RNA Defend Pro (DRDP) (# DRDP_0500, InActiv Blue, Beernem, Belgium) and not the original collection buffers, 0.5 mL of left-over buffer samples were anonymized and 10-fold diluted with DRDP at MLB immediately prior to transport at room temperature to InActiv Blue. 

Of note, swab samples in PBS should be stored at 4 °C or frozen if the test cannot be performed within 24 h [20]. Specimens can be stored in liquid Amies media for up to 72 h at 4 °C [21]. 

### 4.2. Nucleic Acid Amplification Testing

To compensate for variations in storage conditions, all diluted samples that originally tested positive in MLB by Flow (Roche, Diegem, Belgium) or cobas Liat (Roche) were re-analyzed using the cobas Liat at InActiv Blue to generate a baseline value (cobas influenza A/B and RSV # 08160104190 and/or cobas SARS-CoV-2 kit # 09408592190). Cobas Liat runs were performed according to the manufacturer’s instructions using 0.5 mL of 10-fold diluted DRDP samples [22,23].

### 4.3. Rapid Antigen Testing

All samples diluted with DRDP were analyzed using a rapid antigen test (SARS-CoV-2 & I Influenza A/B and RSV Antigen Combo test kit, # 5501788, Fluorecare, Microprofit Biotech, Shenzhen, China). Because of the low pH, DRDP is not directly compatible with rapid antigen tests containing colloidal gold. Therefore, 7 µL of 1 M NaOH was added to 100 µL of the DRDP-diluted samples and homogenized. The entire 107 µL of neutralized DRDP was pipetted onto the lateral flow cartridge.

### 4.4. Extraction-Free Direct RT-qPCR

Direct RT-qPCR was performed using the One Step PrimeScript III RT-qPCR kit (# RR60TW, Takara, Saint-Germain-en-Laye, France) as described in the manual [24]. According to the DRDP instructions for use, the buffer is compatible with direct RT-qPCR if the volume of the sample does not exceed 10% of the total RT-qPCR volume: 5 µL 2× Takara mix + 1 µL primer/probe mix (250 nM of each primer, 125 nM probe, oligonucleotide sequences of SARS-CoV-2, influenza A/B, and RSVA/B to be found in Appendix A) + 3 µL nuclease-free water (InActiv Blue, # NFW_0015) + 1 µL sample. RT-qPCR was performed using a LightCycler II (Roche) in a 96-well plate. The following cycling protocol was used: reverse transcription (52 °C for 5 min, 95 °C for 10 s), followed by 40 cycles (95 °C for 5 s, 60 °C for 30 s). RT-qPCR conditions were described according to the MIQE guidelines [25].

### 4.5. Stability Testing

As the DRDP instructions for use define stability as the maximum time that the delta-Cq remains ≤2 cycles compared to day 0, the inclusion cut-off for a sample in our stability study was a median Cq value of 2 cycles lower than the ad hoc detection limit. This detection limit was defined as the highest median Cq of a sample with all three replicates positive and subtracted by 2 (i.e., 33.34, 33.57, and 31.99, for influenza A, RSV A, and SARS-CoV-2, respectively). From all 41 (57%) samples with a median Cq lower than the cut-off value at time point 0, a random subset of 31 (76%) was selected for stability testing (i.e., 9 SARS-CoV-2, 13 influenza A, and 9 RSV A). Samples were divided into three 100 µL aliquots and subjected to incubation at 4 °C, 20 °C, or 37 °C. Direct RT-qPCR measurements were performed on incubation day 0, 1, 3, and 7.

### 4.6. Statistical Analyses

Data processing, statistical analysis, and graphing were performed using RStudio (version 2023.12.1 build 402), an integrated development environment for R, a programming language for statistical computing and graphics. The following packages were used: cowplot (version 1.1.3), dplyr (version 1.1.4), ggplot2 (version 3.5.0), ggpubr (version 0.6.0), gridExtra (version 2.3), readxl (version 1.4.3), stringr (version 1.5.1), stats (version 4.3.2) and tidyverse (version 2.0.0).

Statistical tests, correlation, *t*-test, and Mann–Whitney were calculated using the “stats” package.

## Figures and Tables

**Figure 1 ijms-25-09097-f001:**
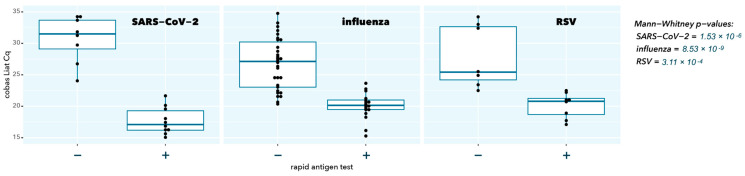
Cobas Liat Cq values stratified according to rapid antigen test result (individual Cq values are solid black dots; boxplot indicates median (horizontal bar), interquartile range (box), and extreme values (vertical line); − means negative RAT, + means positive RAT).

**Figure 2 ijms-25-09097-f002:**
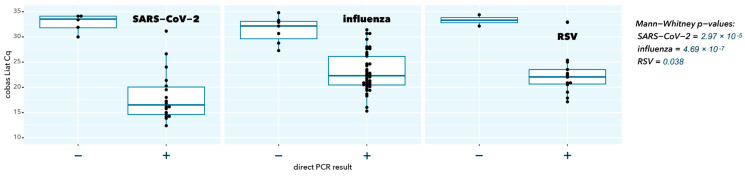
Cobas Liat Cq values stratified according to direct RT-qPCR result (individual Cq values are solid black dots; boxplot indicates median (horizontal bar), interquartile range (box), and extreme values (vertical line); − means negative direct PCR result, + means positive direct PCR result).

**Figure 3 ijms-25-09097-f003:**
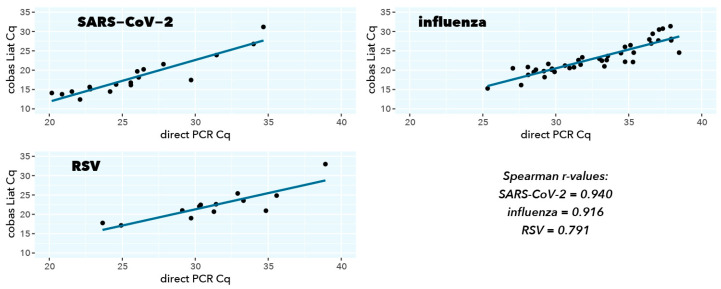
Scatterplot correlation and ordinary least-squares regression of Cq values from direct RT-qPCR (*x*-axis) and cobas Liat (*y*-axis).

**Table 1 ijms-25-09097-t001:** Cross table of 10-fold diluted samples tested by cobas Liat and rapid antigen test.

	Cobas Liat Positive	Cobas Liat Negative	Total
rapid antigen test positive	40	0	40
rapid antigen test negative	48	7	55
total	88	7	95

**Table 2 ijms-25-09097-t002:** Cross table of samples tested by cobas Liat and direct RT-qPCR.

	Cobas Liat Positive	Cobas Liat Negative	Total
direct RT-qPCR positive	72	0	72
direct RT-qPCR negative	16	7	23
total	88	7	95

**Table 3 ijms-25-09097-t003:** Mean delta-Cq, standard error of the mean (SEM), Spearman rank correlation, and *p*-value of Cq value correlation are shown in Figure 3.

	SARS-CoV-2	Influenza A	RSV A
number of samples	19	40	13
mean delta-Cq (SEM)	−7.7 (0.45)	−9.61 (0.28)	−8.92 (0.61)
r (Spearman)	0.940	0.916	0.791
*p* (Spearman)	1.10 × 10^−6^	2.20 × 10^−16^	2.08 × 10^−3^

**Table 4 ijms-25-09097-t004:** Cross table of 31 samples tested by direct RT-qPCR after incubation for 0, 1, 3, and 7 days at 4 °C, 20 °C, and 37 °C.

Samples Positive					
		Day			
Target	Temperature	0	1	3	7
SARS-CoV-2	4 °C	9	9	9	9
	20 °C	9	9	9	9
	37 °C	9	9	9	9
influenza A	4 °C	13	13	13	13
	20 °C	13	13	13	13
	37 °C	13	13	13	13
RSV A	4 °C	9	9	9	9
	20 °C	9	9	9	9
	37 °C	9	9	8	9

**Table 5 ijms-25-09097-t005:** Mean delta-Cq (and standard error of the mean, SEM) of samples after incubation for 1, 3, and 7 days at 4 °C, 20 °C, and 37 °C compared to day 0 (bold indicates conditions with delta-Cq > 2).

Delta-Cq (SEM)					
		Day			
Target	Temperature	0	1	3	7
SARS-CoV-2	4 °C	0.00 (0.00)	0.29 (0.59)	−0.10 (0.53)	1.84 (0.35)
	20 °C	0.00 (0.00)	1.51 (0.62)	1.51 (0.43)	**2.73 (0.58)**
	37 °C	0.00 (0.00)	1.66 (0.71)	1.22 (0.63)	**2.01 (0.62)**
influenza A	4 °C	0.00 (0.00)	0.04 (0.51)	−0.61 (0.49)	0.92 (1.00)
	20 °C	0.00 (0.00)	−1.36 (0.78)	**2.35 (0.43)**	1.09 (0.88)
	37 °C	0.00 (0.00)	0.03 (0.77)	1.19 (0.82)	−0.54 (1.02)
RSV A	4 °C	0.00 (0.00)	0.08 (0.70)	1.00 (0.45)	0.85 (0.69)
	20 °C	0.00 (0.00)	0.88 (0.70)	0.28 (0.90)	**2.51 (0.61)**
	37 °C	0.00 (0.00)	−0.67 (0.95)	0.76 (0.90)	**6.59 (1.96)**

## Data Availability

The raw data supporting the conclusions of this article will be made available by the authors upon request.

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
