# Peer review of "Assessment of DNA/RNA Defend Pro: An Inactivating Sample Collection Buffer for Enhanced Stability, Extraction-Free PCR, and Rapid Antigen Testing of Nasopharyngeal Swab Samples"

_ijms, 2024, doi:10.3390/ijms25169097_

Round 1

Reviewer 1 Report (Previous Reviewer 1)

Comments and Suggestions for Authors

This manuscript has evaluated the DNA/RNA Defend Pro (DRDP) sample collection buffer designed to inactivate and stabilize patient samples. The authors found that DRDP guaranteed viral RNA stability at all temperatures for influenza, SARS-CoV-2, and RSV, showing stability up to 7 days at 4 °C. DRDP-diluted samples exhibited compatibility with direct RT-qPCR and rapid antigen tests.

 Several suggestions:

1.  [nucleic acid amplification testing (NAAT)] in line 24 should put in line 18 at its first appearance.

2.  line 89, full names for [SARS-CoV-2, and RSV] at their first appearances. [influenza] or [influenza A virus]? [RSV] or [RSV A]?

3.   line 148, [NAAT] for [nucleic acid amplification test (NAAT)].

4.   DRDP is an inactivating sample collection buffer. However, no inactivating virus results were shown in this manuscript, but in ref. 12.

5.    Tables 4 and 5 show the RNA stability in DRAP. Whats the RNA stability in PBS or Amies buffer?

Author Response

Comment 1: [nucleic acid amplification testing (NAAT)] in line 24 should put in line 18 at its first appearance.

Response 1: This has been corrected.

Comment 2:  line 89, full names for [SARS-CoV-2, and RSV] at their first appearances. [influenza] or [influenza A virus]? [RSV] or [RSV A]?

Response 2: This has been corrected.

Comment 3: line 148, [NAAT] for [nucleic acid amplification test (NAAT)].

Response 3: This has been corrected.

Comment 4: DRDP is an inactivating sample collection buffer. However, no inactivating virus results were shown in this manuscript, but in ref. 12.

Response 4: Indeed, this manuscript does not deal with the inactivation properties of the DRDP buffer but with direct PCR, RNA stability, and rapid antigen test compatibility. The buffer’s inactivation properties are mentioned in the manufacturer’s Instructions for Use, which includes international guidelines for setting up such studies.

Comment 5: Tables 4 and 5 show the RNA stability in DRAP. What’s the RNA stability in PBS or Amies buffer?

Response 5: We have not evaluated RNA stability in PBS or Amies buffer as these buffers were neither developed nor marketed for RNA stability purposes. Also, this is beyond the scope of our study, in which we evaluate the RNA stabilizing properties of the new DRDP buffer. To enable this, we created a new zero time point (starting time point, t0) when diluting the clinical samples in DRDP buffer. All later time points are compared against time point 0 to evaluate RNA stability of a clinical sample diluted in DRDP.

In reference to https://www.ncbi.nlm.nih.gov/pmc/articles/PMC9924993/, samples in PBS are recommended to be stored at 4 °C or frozen if the test cannot be performed within 24 hours.

Of note, liquid Amies has been developed for preservation of anaerobic bacteria, but is sometimes used to collect and transport specimens for viral studies. Hence, according to the US-FDA, Liquid Amies media may be used for viral transport if universal transport mediium is not available. Specimens can be stored in liquid Amies media for up to 72 hours at 4°C. Reference: https://www.dhs.gov/sites/default/files/publications/2021-hqfo-00070_-_records.pdf

Reviewer 2 Report (Previous Reviewer 3)

Comments and Suggestions for Authors

Dear Editor,

I am writing to provide my assessment following the second review of the manuscript titled "Assessment of DNA/RNA Defend Pro: an inactivating sample collection buffer for enhanced stability, extraction-free PCR, and rapid antigen testing of nasopharyngeal swab samples," authored by Mikhail Claeys, Saif Al Obaidi, Karen Bruyland, Ilse Vandecandelaere, and Jo Vandesompele.

After carefully reviewing the revisions submitted by the authors, I am pleased to inform you that they have addressed all of my previous concerns satisfactorily. The authors have made significant improvements to the manuscript by following the feedback provided, which has greatly enhanced the clarity and scientific rigor of the study.

For example, the authors have reorganized the abstract to ensure consistency in the order of presentation of the results, aligning it with the sequence in the main text. This adjustment improves the coherence of the manuscript, making it easier for readers to follow the study's findings from the abstract through to the detailed results.

Additionally, the authors have made crucial corrections in the figures, such as inverting the axis titles in Figure 3 to maintain consistency across all figures. This seemingly minor adjustment is important for ensuring that the data presentation is uniform and that the readers can accurately interpret the findings. They have also emphasized the Spearman's rank correlation coefficient in the figure, as recommended, which provides a clearer understanding of the association between the variables studied.

Further, the authors have corrected errors identified in the tables, such as the values for SARS-CoV-2 on day 0 across different temperatures and have ensured that all relevant delta Cq values are highlighted as described. These corrections demonstrate the authors' attention to detail and their commitment to improving the accuracy and reliability of their data.

The discussion section has been reordered to follow the logical sequence of the results, thereby strengthening the connection between the findings and their implications. This reorganization ensures that each result is discussed in context, enhancing the overall narrative of the manuscript.

Moreover, the authors have added a conclusion to the manuscript, which was previously missing. This addition provides a succinct summary of the study's contributions and implications, thereby completing the manuscript and reinforcing its significance.

In the Methods section, the authors have included a description of the statistical analyses used, which was previously lacking. This addition is essential for understanding how the data was processed and interpreted, thereby enhancing the scientific rigor of the study.

The authors also moved the methodological details of the stability testing from the Results section to the Methods section, as suggested. This adjustment allows the Results section to focus on presenting the data, while the Methods section now appropriately details the procedures followed, thereby improving the manuscript's overall structure.

In conclusion, I am satisfied with the revisions made by the authors and believe that the manuscript now meets the standards required for publication. The authors have demonstrated their responsiveness to the feedback provided and have improved the manuscript in meaningful ways. I recommend that this manuscript be accepted for publication.

Author Response

Original comments provided to ijms-2942504 were already addressed in this manuscript.

Thank you for your feedback.

Round 2

Reviewer 1 Report (Previous Reviewer 1)

Comments and Suggestions for Authors

The issues I have raised previously have been addressed in this revised manuscript.

Minor suggestions:

The following information mentioned in the authors response should be included in the manuscript to show the enhanced RNA stability of DNA/RNA Defend Pro.

1. samples in PBS should be stored at 4 °C or frozen if the test cannot be performed within 24 hours. [https://www.ncbi.nlm.nih.gov/pmc/articles/PMC9924993/]

2. Specimens can be stored in liquid Amies media for up to 72 hours at 4°C. [ https://www.dhs.gov/sites/default/files/publications/2021-hqfo-00070_-_records.pdf]

Author Response

Reviewer 1

Comments and Suggestions for Authors

The issues I have raised previously have been addressed in this revised manuscript.

Minor suggestions:

The following information mentioned in the author’s response should be included in the manuscript to show the enhanced RNA stability of DNA/RNA Defend Pro.

  1. samples in PBS should be stored at 4 °C or frozen if the test cannot be performed within 24 hours. [https://www.ncbi.nlm.nih.gov/pmc/articles/PMC9924993/]
  2. Specimens can be stored in liquid Amies media for up to 72 hours at 4°C. [ https://www.dhs.gov/sites/default/files/publications/2021-hqfo-00070_-_records.pdf]

Reply

We have added the recommended storage conditions for PBS and liquid Amies to the Sample collection section in Materials and Methods.

This manuscript is a resubmission of an earlier submission. The following is a list of the peer review reports and author responses from that submission.

Round 1

Reviewer 1 Report

Comments and Suggestions for Authors

This article has evaluated the DNA/RNA Defend Pro (DRDP) sample collection buffer designed to inactivate and stabilize patient samples. They found that DRDP guaranteed viral RNA stability at all temperatures for influenza, SARS-CoV-2 and RSV showing stability up to 7 days at 4 °C. DRDP-diluted samples exhibited robust compatibility with direct PCR, as 72 out of 88 samples positive for cobas Liat also tested positive for direct PCR.

 Several suggestions:

1.                line 16 [and others], does [diluted 1/10] mean [10 fold dilution]?

2.                    line 39, please add reference(s) after [at the test laboratory].

3.                    line 87 [and others], [direct PCR] may be changed to [direct RT-PCR], like [RT-qPCR] written in Table 4.

4.                    Table 4, page 5, the number [1] should be [9].

5.      line 204, please add reference(s) after [the manufacturer’s instructions].

6.      line 209, please add reference(s) after [described in the manual].

7.      Many viral transport mediums and publications related to this topic are available now. Please discuss the significance of this study and compare it with previous studies related to viral transport mediums.

Author Response

Dear editor

Thank you for the positive feedback and valuable comments we received on our manuscript. Please find below a point-by-point response. We look forward to the acceptance and subsequent publication of our study.

Reviewer 1

1.line 16 [and others], does [diluted 1/10] mean [10 fold dilution]?

Answer: Yes it does. We have updated the text.

2.line 39, please add reference(s) after [at the test laboratory].

            Answer: We added a reference that documents the feasibility and utility of inactivating infectious samples. Of note, we believe it is common sense and does not need extensive referencing.

  1. line 87 [and others], [direct PCR] may be changed to [direct RT-PCR], like [RT-qPCR] written in Table 4.

            Answer: This has been changed throughout the manuscript.

4.Table 4, page 5, the number [1] should be [9].

            Answer: Corrected.

5.line 204, please add reference(s) after [the manufacturer’s instructions].

Answer: reference or URL were added

6.line 209, please add reference(s) after [described in the manual].

Answer: reference or URL was added

 7.Many viral transport mediums and publications related to this topic are available now. Please discuss the significance of this study and compare it with previous studies related to viral transport mediums.

Answer: A section was added to the Introduction highlighting the uniqueness of our study and      transport medium in the light of what was studied before.

Attached you can find the updated manuscript.

Kind regards,

Mikhail Claeys

Reviewer 2 Report

Comments and Suggestions for Authors

In the present manuscript, the authors evaluate the DNA/RNA Defend Pro (DRDP) sample collection buffer, assessing its efficacy in ensuring sample stability, facilitating extraction-free PCR, and compatibility with rapid antigen testing. Results show DRDP ensures viral RNA stability, compatibility with direct PCR, and potential for optimizing diagnostic processes in resource-limited or time-sensitive scenarios.

In the manuscript, minor issues that should be addressed in order to improve value of this paper.

1.  Authors should consider expanding the introductory paragraph, including additional references

2.  Authors should separate materials and methods e.g. into "sample collection", into "RT-PCR". Furthermore, for each reagent used they should enter the manufacturer and its address.

Author Response

Dear editor

Thank you for the positive feedback and valuable comments we received on our manuscript. Please find below a point-by-point response. We look forward to the acceptance and subsequent publication of our study.

Reviewer 2

In the present manuscript, the authors evaluate the DNA/RNA Defend Pro (DRDP) sample collection buffer, assessing its efficacy in ensuring sample stability, facilitating extraction-free PCR, and compatibility with rapid antigen testing. Results show DRDP ensures viral RNA stability, compatibility with direct PCR, and potential for optimizing diagnostic processes in resource-limited or time-sensitive scenarios.

 In the manuscript, minor issues that should be addressed in order to improve value of this paper.

  1. Authors should consider expanding the introductory paragraph, including additional references

Answer: See Reviewer 1.

  1. Authors should separate materials and methods e.g. into "sample collection", into "RT-PCR". Furthermore, for each reagent used they should enter the manufacturer and its address.

Answer: The materials and methods section was partitioned in different subsections, and               suppliers’ addresses were added.

Attached you can find the updated manuscript.

Kind regards,

Mikhail Claeys

Reviewer 3 Report

Comments and Suggestions for Authors

Dear Editor,

I read with interest the article titled "Assessment of DNA/RNA Defend Pro: an inactivating sample collection buffer for enhanced stability, extraction-free PCR, and rapid antigen testing of nasopharyngeal swab samples" by Mikhail Claeys et al., under review for International Journal of Molecular Sciences in 2024.

The authors have conducted a comprehensive evaluation of the DNA/RNA Defend Pro (DRDP) sample collection buffer, which is designed to inactivate and stabilize patient samples for diagnostic purposes. The primary objectives of the study were to assess the efficacy of DRDP in ensuring sample stability, facilitating extraction-free polymerase chain reaction (PCR), and ensuring compatibility with rapid antigen testing.

In the study conducted by Medisch Labo Bruyland (MLB) and InActiv Blue, ninety-five nasopharyngeal swab samples tested for influenza A/B, RSV A/B, and SARS-CoV-2 were evaluated. Upon retesting a 1/10 dilution using cobas Liat in the InActiv Blue laboratory, 88 out of the 95 samples (93%) initially positive according to the diagnostic test in MLB also tested positive. The samples were diluted and stored at varying temperatures and for different durations, leading to an expectation that a retest might not detect the samples with the lowest signals.

Of the 88 samples that tested positive on cobas Liat, 40 (45%) were positive for rapid antigen testing. The 7 samples that tested negative on cobas Liat were also negative for rapid antigen testing, with no false positive results obtained. Samples with a negative rapid antigen test had a significantly higher cobas Liat Cq value, confirming the superior analytical sensitivity of a molecular test.

Furthermore, of the 88 samples positive for cobas Liat, 72 (82%) were positive for direct PCR. The 7 samples that tested negative for cobas Liat also tested negative for direct PCR, with no false positive results obtained. Samples with a negative direct PCR result had a significantly higher cobas Liat Cq value, in line with the higher sensitivity due to a 200-fold higher input in cobas Liat testing compared to direct PCR.

A good to excellent correlation was observed between the direct PCR test and the CE-IVDR cobas Liat test for all viral targets. The mean differences in Cq value between the direct PCR and cobas Liat were closely in line with the approximately 200-fold difference in input.

For stability testing, a semi-random subset of 31 out of the 72 samples that tested positive with direct PCR and cobas Liat was selected. The DRDP instructions for use define stability as the maximum time that the delta-Cq remains ≤ 2 cycles compared to day 0. All targets in all samples were detected on day 7 at all temperatures. Influenza samples remained stable for up to 7 days at all tested temperatures, while SARS-CoV-2 and RSV A samples remained stable up to 3 days at all tested temperatures and up to 7 days at 4 °C.

In the discussion and conclusion of the study, the authors highlight the effectiveness of the DNA/RNA Defend Pro (DRDP) sample collection buffer in inactivating pathogens, stabilizing viral RNA, and facilitating extraction-free polymerase chain reaction (PCR) and rapid antigen testing. 

The study demonstrates that DRDP ensures viral RNA stability at varying temperatures and is compatible with different diagnostic tests, including direct PCR and rapid antigen testing. 

Although direct NAAT showed somewhat lower sensitivity compared to CE-IVDR NAAT, this is attributed to the dilution factor and the smaller sample volume used in direct NAAT. The DRDP buffer also preserves antigens in their native state, maintaining perfect specificity in rapid antigen testing.

The stability of viral RNA in DRDP is virus-dependent, with recommended storage at 4 °C for longer stability. 

Overall, the study concludes that DRDP is a promising tool for optimizing diagnostic processes, particularly in resource-limited or time-sensitive scenarios, by reducing the need for nucleic acid purification and maintaining the integrity of samples for accurate testing.

Review

The study evaluates a product with the potential to prevent pre-analytical errors in molecular diagnostics by preserving the sample and allowing the execution of multiple tests. It is based on the comparison of laboratory methods. 

Below are my minor revisions with the intention of improving the scientific description of the work and the connection between results, discussion, and methods.

Abstract

The organization of the abstract could benefit from maintaining the order of presentation of the results. The abstract currently presents results on stability, PCR, and then rapid antigen tests, whereas the order of the results in the main text is the characterization of the samples, retesting, rapid antigen test, PCRs, and stability. Ensuring consistency in the order of presentation between the abstract and the main text could enhance the clarity and coherence of the report.

Introduction

The introduction is well-presented and contains a clear objective.

Results

In Figure 3 of the results, it is recommended to invert the titles of the axes (Cobas Liat Cq and Direct PCR Cq) to maintain consistency with Figures 1 and 2. This adjustment will help ensure that the presentation of the data is uniform across all figures, making it easier for readers to interpret and compare the results.

In the presentation of Figure 3, it might be more beneficial to emphasize the Spearman's rank correlation coefficient (r) rather than the p-value (p) associated with the Spearman's test. The correlation coefficient provides a measure of the strength and direction of the association between the two variables, which can be more informative in the context of this type of scatterplot graph. If deemed relevant, consider adjusting the figure or the accompanying text to highlight the Spearman's r value to give readers a clearer understanding of the relationship between Cobas Liat Cq and Direct PCR Cq.

In Table 4, it is important to verify if the values for SARS-CoV-2 on day 0 are indeed 1 for all temperatures (4°C, 20°C, and 37°C).

In Table 5, please ensure that all delta Cq values greater than 2 are highlighted in bold, as described in the accompanying text.

Discussion

I recommend aligning the discussion topics with the sequence of results presented in the results section. This approach would ensure a clear correspondence between the results and their discussion, making it easier for readers to follow the logical flow of the study. Discussing each topic in order allows for a focused examination of each result and its implications, both practical and theoretical. By improving the organization of the discussion, you can enhance the clarity and depth of your analysis, ultimately strengthening the overall coherence of the manuscript.

Conclusion

I noticed that the manuscript currently lacks a conclusion

Methods

The Methods section does not contain a description of any statistical analyses used in the study. It would be beneficial to describe the analyses applied to which variables and the software used for these analyses. This addition would provide clarity on how the data was processed and interpreted, enhancing the scientific rigor of the study.

The stability study should be better described and include methodological information that is currently in the results section (lines 116-121). This part of the results should be appropriately adjusted to focus on the presentation of the obtained data. By incorporating the methodological details into the Methods section, the Results section can be streamlined to emphasize the findings of the study more clearly.

Author Response

Dear editor

Thank you for the positive feedback and valuable comments we received on our manuscript. Please find below a point-by-point response. We look forward to the acceptance and subsequent publication of our study.

Reviewer 3

Below are my minor revisions with the intention of improving the scientific description of the work and the connection between results, discussion, and methods.

Abstract

The organization of the abstract could benefit from maintaining the order of presentation of the results. The abstract currently presents results on stability, PCR, and then rapid antigen tests, whereas the order of the results in the main text is the characterization of the samples, retesting, rapid antigen test, PCRs, and stability. Ensuring consistency in the order of presentation between the abstract and the main text could enhance the clarity and coherence of the report.

Answer: The abstract has been rewritten to reflect the order of the Results section.

Results

In Figure 3 of the results, it is recommended to invert the titles of the axes (Cobas Liat Cq and Direct PCR Cq) to maintain consistency with Figures 1 and 2. This adjustment will help ensure that the presentation of the data is uniform across all figures, making it easier for readers to interpret and compare the results.

Answer: The Figure 3 axes were inverted.

In the presentation of Figure 3, it might be more beneficial to emphasize the Spearman's rank correlation coefficient (r) rather than the p-value (p) associated with the Spearman's test. The correlation coefficient provides a measure of the strength and direction of the association between the two variables, which can be more informative in the context of this type of scatterplot graph. If deemed relevant, consider adjusting the figure or the accompanying text to highlight the Spearman's r value to give readers a clearer understanding of the relationship between Cobas Liat Cq and Direct PCR Cq.

Answer: The Spearman's rank correlation coefficient values were added to the Figure 3             panels. The p-values were only retained in the text.

In Table 4, it is important to verify if the values for SARS-CoV-2 on day 0 are indeed 1 for all temperatures (4°C, 20°C, and 37°C).

Answer: This was an error. Thank you for pointing this out. See also Reviewer 1.

In Table 5, please ensure that all delta Cq values greater than 2 are highlighted in bold, as described in the accompanying text.

Answer: This has been corrected.

Discussion

I recommend aligning the discussion topics with the sequence of results presented in the results section. This approach would ensure a clear correspondence between the results and their discussion, making it easier for readers to follow the logical flow of the study. Discussing each topic in order allows for a focused examination of each result and its implications, both practical and theoretical. By improving the organization of the discussion, you can enhance the clarity and depth of your analysis, ultimately strengthening the overall coherence of the manuscript.

Answer: The Discussion topics were reordered to reflect the order of the Results section.

Conclusion

I noticed that the manuscript currently lacks a conclusion

Answer: A concluding paragraph was added.

Methods

The Methods section does not contain a description of any statistical analyses used in the study. It would be beneficial to describe the analyses applied to which variables and the software used for these analyses. This addition would provide clarity on how the data was processed and interpreted, enhancing the scientific rigor of the study.

Answer: A paragraph on statistical analyses was added to the Methods section.

The stability study should be better described and include methodological information that is currently in the results section (lines 116-121). This part of the results should be appropriately adjusted to focus on the presentation of the obtained data. By incorporating the methodological details into the Methods section, the Results section can be streamlined to emphasize the findings of the study more clearly.

Answer: The methodological part of the stability testing was moved to the Methods section.

Attached you can find the updated manuscript.

Kind regards,

Mikhail Claeys
